# Modulation of Cystatin F in Human Macrophages Impacts Cathepsin-Driven Killing of Multidrug-Resistant *Mycobacterium tuberculosis*

**DOI:** 10.3390/microorganisms11071861

**Published:** 2023-07-24

**Authors:** Manoj Mandal, David Pires, Maria João Catalão, José Miguel Azevedo-Pereira, Elsa Anes

**Affiliations:** 1Host-Pathogen Interactions Unit, Research Institute for Medicines, iMed-ULisboa, Faculty of Pharmacy, Universidade de Lisboa, Av. Prof. Gama Pinto, 1649-003 Lisboa, Portugal; 2Center for Interdisciplinary Research in Health, Católica Medical School, Universidade Católica Portuguesa, Estrada Octávio Pato, 2635-631 Rio de Mouro, Portugal

**Keywords:** tuberculosis, multidrug-resistant TB, cathepsins, cystatins, host-directed therapies

## Abstract

Tuberculosis (TB) treatment relies primarily on 70-year-old drugs, and prophylaxis suffers from the lack of an effective vaccine. Among the 10 million people exhibiting disease symptoms yearly, 450,000 have multidrug or extensively drug-resistant (MDR or XDR) TB. A greater understanding of host and pathogen interactions will lead to new therapeutic interventions for TB eradication. One of the strategies will be to target the host for better immune bactericidal responses against the TB causative agent *Mycobacterium tuberculosis* (Mtb). Cathepsins are promising targets due to their manipulation of Mtb with consequences such as decreased proteolytic activity and improved pathogen survival in macrophages. We recently demonstrated that we could overcome this enzymatic blockade by manipulating protease inhibitors such as cystatins. Here, we investigate the role of cystatin F, an inhibitor that we showed previously to be strongly upregulated during Mtb infection. Our results indicate that the silencing of cystatin F using siRNA increase the proteolytic activity of cathepsins S, L, and B, significantly impacting pathogen intracellular killing in macrophages. Taken together, these indicate the targeting of cystatin F as a potential adjuvant therapy for TB, including MDR and XDR-TB.

## 1. Introduction

Human tuberculosis (TB) caused by the species *Mycobacterium tuberculosis* (Mtb) represents one of the deadliest infections in the world [1]. Mtb is a highly successful pathogen that primarily affects the lungs, leading to pulmonary TB, a condition required for transmission [1,2,3,4,5,6,7,8]. The infection of other organs and tissues may also occur during extrapulmonary TB but constitutes a dead end for the pathogen [3,4,9]. Approximately 25% of the world’s population has an immunologic activation status indicative of prior infection with Mtb while not presenting disease symptoms [1,3,10,11]. In this latent infection, Mtb remain quiescent for decades until the reactivation of the infection, a phenomenon that will result in disease (active TB) in 5 to 10% of latent cases [1]. The most recent WHO report indicates that in 2021 ten million people developed active TB, with 1.6 million deaths. Among the active TB cases, 450,000 were caused by Mtb strains with multidrug (MDR) or extensive (XDR) drug resistance to available antibiotics. First-line chemotherapy includes the four first-line antibiotics rifampicin, isoniazid, pyrazinamide, and ethambutol for the first two months and an extension for four months with rifampicin and isoniazid [12]. With the emergence of MDR (resistant to isoniazid and rifampicin) and XDR (MDR with resistance to any fluoroquinolone and one of three injectable second-line drugs) TB, treatment regimens are more complicated and lengthier (lasting for up to two years) with the inclusion of alternative less efficient second-line agents and high rates of failure [1,13,14]. The accelerated approval of clinical trials that are still in phase three for delamanid, bedaquiline, and pretomanid for complicated cases of MDR/XDR alone allows a limited treatment time but toxic issues and newly revealed drug resistance are still a concern [15,16,17].

The public health threats of MDR and XDR-TB call for innovative therapeutic approaches, including host-directed therapies that can also be applied as adjunctive anti-microbial treatment.

Professional phagocytes such as macrophages are archetype cells for pathogen destruction yet paradoxically contribute to a successful Mtb infection [18]. They contribute to the establishment of Mtb intracellular niches in early phagosomes [2,19] and to the expansion of infected cells in the lung parenchyma in structures typical of TB: the granuloma [2,20,21,22]. Finally, macrophages provide means for the induction of an exacerbated inflammatory response required for necrosis, a condition concomitant with uncontrolled Mtb replication [23,24,25]. The consequent lung cavitation will lead to pathogen transmission.

Endolysosomal cathepsins in macrophages play relevant innate and adaptive immune responses to control pathogens, including Mtb [3,26]. They are directly involved in phagolysosomal pathogen proteolytic digestion and killing while making bridges for adaptive T- lymphocyte activation [19,27,28,29].

Our group provided evidence of Mtb’s ability to manipulate lysosomal cathepsins and their natural inhibitors, cystatins [28,29,30], leading to their intracellular survival and poor T cell activation. Furthermore, particularly for cystatins, we found a significantly increased gene expression in the early stages of infection for cystatins C, A, and SN common to infection with Mtb and HIV [30].

We overcame the Mtb-induced inhibition of cathepsins’ proteolytic activity using different approaches such as targeting the regulation of gene expression using microRNAs or repurposing the HIV protease inhibitor saquinavir [29,31,32,33] that we demonstrate to activate cathepsins B, L, and S. One very successful attempt to reverse the cathepsins’ manipulation using Mtb was achieved by targeting cystatin C using RNA silencing [30,34].

During those gene expression studies, cystatin F (CstF) gene (CST7) emerged as the most upregulated gene during mycobacteria infection of human macrophages, including Mtb and the non-pathogen saprophyte species *Mycobacterium smegmatis* [30].

Indeed, cystatin F is among the most upregulated transcripts in dendritic cells activated by bacterial lipopolysaccharide [35] and was described to be downregulated in all-trans retinoic acid (ATRA)-stimulated U937 cells (causing monocytic differentiation towards macrophages) [36]. However, because there are almost no reports on this cathepsin inhibitor during Mtb infection of human macrophages, in this work, we aim to decipher their role in Mtb intracellular killing, including during infection with MDR and XDR clinical Mtb strains.

## 2. Materials and Methods

### 2.1. Cell Isolation and Culture Conditions

Primary human monocyte-derived macrophages were isolated and then differentiated from buffy coats of healthy human donors, which were provided by the National Blood Institute (Instituto Português do Sangue e da Transplantação, I.P., Lisbon, Portugal) following a previously described protocol [30].

### 2.2. Bacterial Cultures

*M. tuberculosis* H37Rv (ATCC 27294) (American Type Culture Collection, Manassas, VA, USA) and clinical strains provided and characterized by the TB National Reference Laboratory from the Portuguese National Institute of Health Dr. Ricardo Jorge (INSA) were grown in Middlebrook’s 7H9 medium supplemented with 10% OADC enrichment (BD Difco, Franklin Lakes, NJ, USA), 0.02% glycerol, and 0.05% tyloxapol (Merck, KGaA, Darmstadt, Germany) at 37 °C. The clinical strain (INSA code 33427) is susceptible to streptomycin, isoniazid, rifampicin, and pyrazinamide; the MDR strain (INSA code 34192) is resistant to all those antibiotics in addition to ethionamide. The XDR strain (INSA code 163761) is resistant to all of the previous named antibiotics, and, additionally, to amikacin, kanamycin, capreomycin, moxifloxacin, and ofloxacin. All experimental procedures using live Mtb strains were performed in the biosafety level 3 laboratory at the Faculty of Pharmacy of the University of Lisbon, maintaining the national and European containment level 3 laboratory management and biosecurity standards based on applicable EU directives. The faculty’s biological safety committee has also approved all those procedures.

### 2.3. Macrophage Infection

Before infection, all Mtb strains were cultivated for approximately seven days at 37 °C and 5% CO_2_ until the exponential growth phase was reached. On the day of infection, the bacterial suspensions were centrifuged and washed in phosphate-buffered saline (PBS) and resuspended in macrophage culture medium without antibiotics. Clumps of bacteria in the suspension were disrupted using an ultrasonic bath treatment for 5 min and removed using centrifugation at a low speed of 500× *g* for 1 min. The obtained single-cell suspension was verified through fluorescence microscopy and quantified by measuring the optical density at 600 nm. Then, the infection was performed with a multiplicity of infection (MOI) of 1 bacterium per macrophage for 3 h at 37 °C and 5% CO_2_. Following this incubation period, the cells were washed with PBS to remove free bacteria and with an added fresh, complete medium.

### 2.4. Transfection

Macrophages were transfected 72 h before infection to achieve maximum silencing. Transfection with anti-CstF siRNA or with scramble control siRNA was performed using ScreenFect A (ScreenFect GmbH, Eggenstein-Leopoldshafen, Germany) transfection reagent according to the manufacturer’s protocol. Macrophages were incubated for 72 h with the transfection reagent and SMARTpool ON-TARGETplus Human CST7 siRNA (Agilent Technologies, Inc., Santa Clara, CA, USA); target sequences: AGUGAAAGGCCUGAAAUAU, GAAAUUGGCAGAACUACCU, GGAUGACUGUGACUUCCAA, and CAAGGGCCCUAGUUCAGAU) or the respective siRNA non-targeting (scramble) control (Agilent Technologies, Inc., Santa Clara, CA, USA); target sequences: UGGUUUACAUGUCGACUAA, UGGUUUACAUGUUGUGUGA, UGGUUUACAUGUUUUCUGA, and UGGUUUACAUGUUUUCCUA) in the medium without an antibiotic. Silencing efficacy was evaluated using qPCR data analysis.

### 2.5. Macrophage Viability

Macrophages after transfection with anti-CstF siRNA or with scramble control siRNA were incubated with 10% (V/C) PrestoBlue (Invitrogen, Carlsbad, CA, USA) resazurin-based solution at 37 °C and 5% CO_2_. After 2–3 h of incubation, fluorescence emission was analyzed according to the manufacturer’s instructions using a Varioskan™ LUX Multimode Microplate Reader (Thermo Fisher Scientific, Waltham, MA, USA). Non-transfected macrophages were used as a reference for 100% viability, whereas macrophages treated with 0.05% Igepal were used as a reference for 0% viability.

### 2.6. Reverse Transcriptase-qPCR

RNA was isolated from macrophages using NZY Total RNA Isolation kit (NZYTech, Lisbon, Portugal) following the manufacturer’s instruction. Total RNA measuring 100 ng was used for cDNA synthesis using an NZY First-Strand cDNA Synthesis Kit (NZYTech, Lisbon, Portugal) according to the instructions provided by the manufacturer. To perform qPCR, an NZY qPCR Green Master Mix (NZYTech, Lisbon, Portugal) with a primer set specifically for CST7 mRNA (Forward-TCCCCAGATACTTGTTCCCAGG; Reverse-TTCTGCCAATTTCCACCTCCA) and for glyceraldehyde 3-phosphate dehydrogenase (GAPDH) mRNA (Forward-AAGGTGAAGGTCGGAGTCAA; Reverse-AATGAAGGGGTCATTGATGG) was used proceeding according to the previously described conditions [30]. The qPCR was performed using a QuantStudio^TM^ 7 Flex System (De Novo Software, Pasadena, CA, USA) (Thermo Fisher Scientific, Waltham, MA, USA), and data analysis was carried out using the ΔΔCt method. The mRNA expression profiles were normalized to the GAPDH housekeeping gene and calculated relative to the samples treated with the scramble control siRNA.

### 2.7. Western Blotting

Proteins were harvested using a RIPA buffer (Merck, KGaA, Darmstadt, Germany). All the protein samples were diluted 1:1 in Laemmli buffer (Merck, KGaA, Darmstadt, Germany) and heated at 95 °C for 5 min before running the gel. Proteins were separated using sodium dodecyl sulfate–polyacrylamide gel electrophoresis (SDS–PAGE) using 4–20% Mini-PROTEAN TGX Precast Protein Gels (Bio-Rad Laboratories, Hercules, CA, USA). They were transferred to the nitrocellulose membrane through the Trans-Blot Turbo Transfer System (Bio-Rad Laboratories, Hercules, CA, USA). Next, the membrane was processed and stained using the iBind Western system (Thermo Fisher Scientific, Waltham, MA, USA) and primary antibodies specific for CstF (1:2000 dilution of #SAB2700222, Sigma-Aldrich), ß-tubulin (1:4000 dilution of #ab6046, Abcam, Cambridge, UK), and horseradish peroxidase (HRP)-conjugated secondary antibody (1:4000 dilution of #1706515, Bio-Rad Laboratories, Hercules, CA, USA). The visualization of bands was performed through chemiluminescence using an NZY Supreme ECL HRP Substrate (NZYTech, Lisbon, Portugal) in a ChemiDoc XRS+ System (Bio-Rad Laboratories, Hercules, CA, USA). The band intensity was quantified using ImageLab software on USB drive #12012931 (Bio-Rad Laboratories, Hercules, CA, USA).

### 2.8. Flow Cytometry

Macrophages were infected for three hours with a GFP-expressing Mtb strain to quantify bacterial internalization or with non-fluorescent H37Rv and clinical strains to quantify cell death. Macrophages were washed with PBS following three hours of infection and immediately detached (internalization experiments) or incubated for an additional three days (cell death experiments). Infected macrophages were detached using accutase. For cell death staining, Apotracker Green and Zombie Red (Biolegend, San Diego, CA, USA) dyes were used to stain apoptotic and dead cells, respectively. Furthermore, they were fixed in 4% paraformaldehyde for an hour and then analyzed in a Cytek^®^ Aurora flow cytometer (Cytek^®^ Biosciences, Fremont, CA, USA). Data analysis was performed in FCS Express 7 (De Novo Software, Pasadena, CA, USA).

### 2.9. Bacterial Intracellular Survival: Colony-Forming Unit Assay

Infected macrophages at the selected time points (T_0_, T_1_, T_3_, and T_6_) of infection were lysed to recover intracellular bacteria using 0.05%of Igepal solution for 15 min. The resulting bacterial suspensions were serially diluted and plated in Middlebrook’s 7H10 solid medium with 10% OADC (BD Difco, Franklin Lakes, NJ, USA), and incubated for 2–3 weeks at 37 °C and 5% CO_2_ before colonies were observable and able to be counted under a microscope.

### 2.10. Enzymatic Activities of Cathepsins

After 24 h of infection, macrophages in 96-well plates were lysed with chilled cathepsin-specific lysis buffer and incubated on ice for 10 min. Cells were centrifuged at maximum speed for 5 min to recover the supernatant and a reaction buffer and substrate provided in the kit for cathepsin B or for cathepsin L (Abnova™; Thermo Fisher Scientific, Waltham, MA, USA), or cathepsin S (Biovision/Abcam, Cambridge, UK) were also added. Cathepsin-specific inhibitor supplied in the kit was used as a control to verify the assay specificity. The mixture was incubated at 37 °C for 1–2 h in a Tecan M200 spectrofluorometer (Tecan Group, Männedorf, Switzerland), taking fluorescence readings every 5 min.

### 2.11. Statistical Analysis

Statistical analysis was performed in GraphPad Prism 9. Multiple group comparisons were conducted using one-way ANOVA followed by a Holm–Sidak post hoc test. Two group comparisons were made using Student’s *t*-test. Differences were considered significant when the calculated adjusted *p* value was equal to or below the alpha level of 0.05.

## 3. Results

### 3.1. siRNA-Mediated Gene Silencing Effectively Lowers Cystatin F Expression in Human Primary Macrophages without Cytotoxic Effects

The manipulation of cystatin F (CstF) using siRNA-mediated gene silencing was performed, and the extension of gene expression silencing efficiency and cytotoxic effects was evaluated in macrophages. The transfection with anti-CstF was compared to a control of scramble siRNA, as previously described in the Materials and Methods section. Results indicate approximately 50% silencing of CstF mRNA (Figure 1a) and 60% of the protein amounts (Figure 1b). Next, the extent of silenced macrophages during infection with Mtb was investigated. From the qPCR data analysis, it was found that 3 h and 24 h post-infection with Mtb, CstF was silenced by approximately 60%. Moreover, after 72 h of infection, CstF mRNA remained at low levels of gene expression (Figure 1c). The results justify the efficacy of siRNA-mediated silencing in human macrophages pre- and post-infection with Mtb. Notably, there was no difference in cell viability when comparing CstF-silenced macrophages to cells transfected with a non-specific RNA (scramble) (Figure 1d).

### 3.2. Silencing of CstF Expression Improves the Intracellular Killing of Mtb Including Clinical Strains with Distinct Drug Resistance Profiles

CstF is primarily expressed in the immune cells, most prominently in dendritic cells, T cells, and NK cells [37]. We decided to investigate the expression of this endogenous protein inhibitor in primary human macrophages, specifically during Mtb infection. Therefore, we targeted CstF through siRNA-mediated gene silencing as previously established in primary human macrophages [38]. To evaluate the impact of CstF silencing on macrophages’ response to Mtb infection, the quantification of the intracellular survival of Mtb was performed through colony forming units (CFU) counts of bacteria recovered from infected cells over six days (Figure 2a). The results indicate a significant impact on the macrophages’ killing ability towards all Mtb strains, whether the Mtb reference laboratory or clinical strains with different drug resistance profiles. Figure 2a,b show a significant reduction in CFU from bacteria recovered from CstF-silenced infected macrophages compared to scramble controls (*p* < 0.001). To assess if the effects on CFU counts were not affected by differences on the ability of macrophages to internalize bacteria, the amounts of Mtb in CstF-silenced macrophages was compared to those in the scramble control using flow cytometry. The results indicate similar amounts of bacteria in both conditions (Figure 2c). Moreover, the results, indeed, show no interference of cell death when comparing clinical and reference strains that could impact differences on CFU counts (Figure 2d). Overall, these results together with previous results (Figure 1d) showing no cytotoxic effects due to CystF silencing, indicate that the modulation of CstF expression in macrophages significantly impacts the intracellular killing of Mtb infection.

### 3.3. Silencing of CstF Expression Significantly Impacts Cysteine Cathepsin Enzymatic Activity in Macrophages Infected with Mtb

The modulation of CstF expression impacts the intracellular survival of Mtb in human macrophages during infection. To investigate whether the intracellular killing of Mtb was attributed to an effect on cathepsins, the proteolytic activity of cathepsins was evaluated. The activity of several cathepsins, including cathepsins B, L, and S, was measured over one hour starting at 24 h post-infection using fluorogenic peptide substrates specific for those cathepsins. Cathepsin-specific inhibitors were used as negative controls. As expected, there was a decrease in the proteolytic activity of cathepsins after Mtb infection, as shown when comparing the activity in scramble non-infected cells with Mtb scramble (Figure 3a–c). In all infected conditions, the depletion of CstF leads to an increased proteolytic activity of cathepsins (Figure 3a–c). Comparatively, CstF silencing does not affect cathepsin activities in non-infected cells. 

## 4. Discussion

Proteolytic enzymes participate in several important physiological processes that maintain host cell homeostasis [39], one of which is the digestion of bacteria uptaken by phagocytosis [40]. Mtb are intracellular pathogens that have their main niche in the phagosomes of host phagocytes such as macrophages. To establish themselves in this niche, the bacilli impair phagosomal maturation and the ensuing intracellular bactericidal mechanisms deployed by the macrophage [19,41]. Our previous works have established how this Mtb-specific manipulation of the host macrophages interferes with their ability to express proteolytic enzymes such as lysosomal cathepsins and how the consequent defect in their activity leads to an improved Mtb intracellular survival [28]. We have thus searched for different strategies to restore cellular proteolytic activity and increase the macrophages’ ability to control the infection.

Cystatins are endogenous regulators of cathepsins and thus represent potential targets to be manipulated to restore cathepsin activity during infection. To this end, we have recently explored the manipulation of cystatin C and successfully developed a macrophage-directed solution that restores cathepsin activity and increases the intracellular killing of Mtb by targeting cystatin C expression [30,34]. Our previous choice for cystatin C was related to its abundant expression and strong inhibitory effect on the most relevant lysosomal cathepsins, such as B, L, and S. In the present work, we have analyzed the potential of a different cystatin, cystatin F, to manipulate the macrophages’ proteolytic activity and improve Mtb killing. Our previous evidence had shown cystatin F to be highly expressed during Mtb infection of primary human macrophages [30]. Yet, contrary to cystatin C, which is ubiquitously expressed and mainly functional in the extracellular milieu [36,39,40], cystatin F is expressed uniquely in immune cells [36,42,43]. Moreover, while potentially secreted out of cells, cystatin F is mainly concentrated in the endolysosomal pathway on the producer cell or nearby immune cells that internalize this cystatin via the Man-6-phosphate receptor [36,42,43,44]. This makes cystatin F a more promising target for the specific regulation of Mtb-infected immune cells with potentially fewer side effects.

Accumulation of cystatin F in endosomal/lysosomal vesicles directly regulates the activity of intracellular cysteine cathepsins [45]. The monomeric form of cystatin F was shown to highly inhibit cathepsins L, V, K, and F, with a lesser extent of inhibition for cathepsins S and H and no effect on cathepsins B and C [37,46]. However, a post-transcriptionally processed form of cystatin F was shown to efficiently inhibit cathepsin C with consequences to the proteolytic activation of serine proteases in T cells, NK cells, neutrophils, and mast cells [42]. The consequent non-activated proteases include granzymes A and B, cathepsin G, elastase, proteinase 3, and mast cell chymase [47].

As mentioned before, cystatin F is highly expressed in dendritic cells, particularly in those undergoing LPS-induced maturation [35]. On the other hand, in the monocytic cell line U937, cystatin F downregulation has been described in response to phorbol ester stimuli inducing differentiation towards macrophages [36].

Here, for the first time, we showed that interference in cystatin F expression in primary human macrophages impacts the expression of cathepsins B, L, and S during Mtb infection, demonstrating a role for this cystatin in regulating these lysosomal cathepsins in macrophages. Per our previous results [28,30,32], when macrophages were infected with Mtb, a concomitant decrease in cathepsin activity was observed for the three cathepsins analyzed. By targeting cystatin F expression, the levels of proteolytic activity could be restored to the basal levels detected before infection. For cathepsins L and S, these results agree with previous reports showing direct regulation using cystatin F [37,46], whereas no evidence of direct inhibition was found for cathepsin B. Our results do not distinguish between a direct or indirect role for cystatin F, which might explain the contradictory results.

As previously mentioned, reports on cystatin F’s role in immune system cells have been primarily focused on its negative regulation of cathepsin-mediated activation of granzymes and the cytotoxic activity of cytotoxic T cells, NK cells, and granulocytes [48]. Here, we demonstrated the impact of cystatin F on primary macrophages’ ability to control Mtb’s intracellular burden. Silencing cystatin F resulted in a lower intracellular survival of the bacteria. Furthermore, we could reproduce those results in several clinical strains with different drug resistance phenotypes. This is the first report connecting cystatin F and a bacterial infection in macrophages. It agrees with the hypothesis that Mtb interferes with macrophages’ cathepsins and that we can restore their function by silencing cathepsin inhibitors, thus improving Mtb killing. This is in line with our previous finding for a different cathepsin inhibitor, cystatin C, also producing similar results in macrophages [30].

Altogether, our results reveal a promising new host molecule that can be targeted to improve the control of Mtb infection, even in cases of multiple and extensive drug resistance. Being a more specific cystatin of immune cells, future studies should explore if other aspects of the immunopathology induced by Mtb can be interfered with by targeting this cystatin.

## Figures and Tables

**Figure 1 microorganisms-11-01861-f001:**
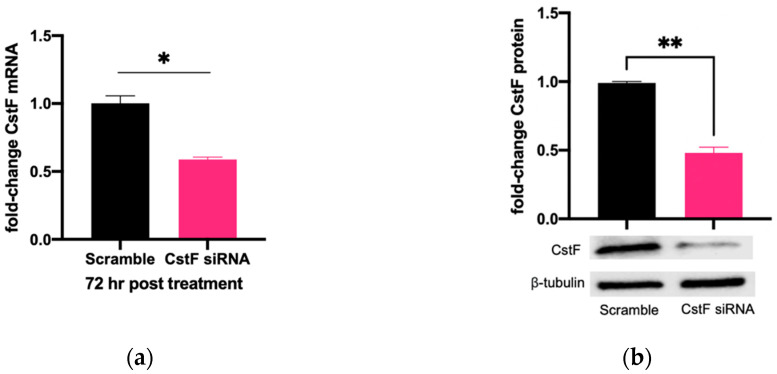
siRNA-mediated gene silencing effectively decreases CstF expression in human macrophages. To achieve maximum silencing, CstF was silenced by siRNA 3 days prior to infection with Mtb. (**a**) Relative gene expression of CstF mRNA in macrophages was obtained using RT-qPCR after 72 h of transfection. (**b**) The Western blot image demonstrates the silencing of the CstF protein by siRNA at the moment of infection. The respective bar plot was calculated from two independent experiments measuring band intensity using β-tubulin as a calibrator. The error bars represent the standard error of the mean. (**c**) RT-qPCR measured the relative gene expression of CstF mRNA in transfected cells 3 h, 24 h, and 72 h post-infection. The bar plots represent the average of three independent experiments, and the error bars demonstrate the standard error of the mean. (**d**) Effects of silencing on the viability of cells transfected with siRNA for CstF relative to scramble transfected cells. This was measured using PrestoBlue (resazurin-based solution) and quantifying the fluorescence emission in a plate reader. Untreated macrophages with 100% viability and 0.05% Igepal-treated macrophages with 0% viability were used as controls. * *p* < 0.05, ** *p* < 0.01.

**Figure 2 microorganisms-11-01861-f002:**
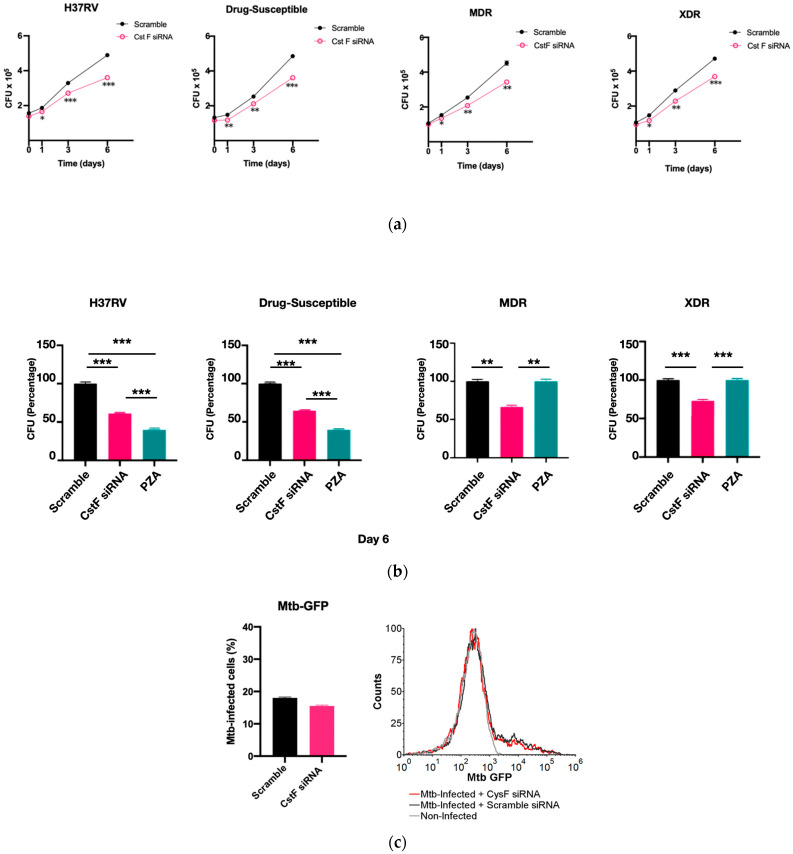
Silencing of CstF expression results in increased intracellular killing of reference laboratory and clinical strains of Mtb with distinct drug resistance profiles in primary human macrophages. Colony-forming units (CFU) of intracellular bacteria were recovered from human monocyte-derived macrophages transfected with anti-CstF siRNA or scramble control siRNA 72 h before infection with (**a**) Mtb reference laboratory strain H37RV and clinical strains. The intracellular survival of bacteria was evaluated at different time points. The line plots depict mean CFU per sample. (**b**) The bar plots demonstrate average CFU compared to the respective controls at Day 6 post-infection. The error bars represent the standard error of the mean. Pyrazinamide (PZA) was used as a reference to determine the killing efficacy. (**c**) The percentage of Mtb-infected macrophages was determined using flow cytometry in the scramble and CstF-silenced infected cells using a GFP-expressing Mtb strain (H37Rv). The bar plots represent the average of three biological replicates, while the error bars depict the standard error. The fluorescence intensity histograms present raw values from one of the replicates. (**d**) Cell death quantified using flow cytometry by staining macrophages with Apotracker Green (apoptosis) and Zombie Red (dead cells) dyes following 3 days of infection with laboratory and clinical strains. Live apoptotic cells were considered “early apoptotic”, dead apoptotic cells were considered “late apoptotic”, while non-apoptotic dead cells were considered “necrotic”. The bar plot represents the mean of two independent experiments and the error bars represent the standard error of the mean. The dot plots depict representative results from one experimental replicate. * *p* < 0.05, ** *p* < 0.01, *** *p* < 0.001.

**Figure 3 microorganisms-11-01861-f003:**
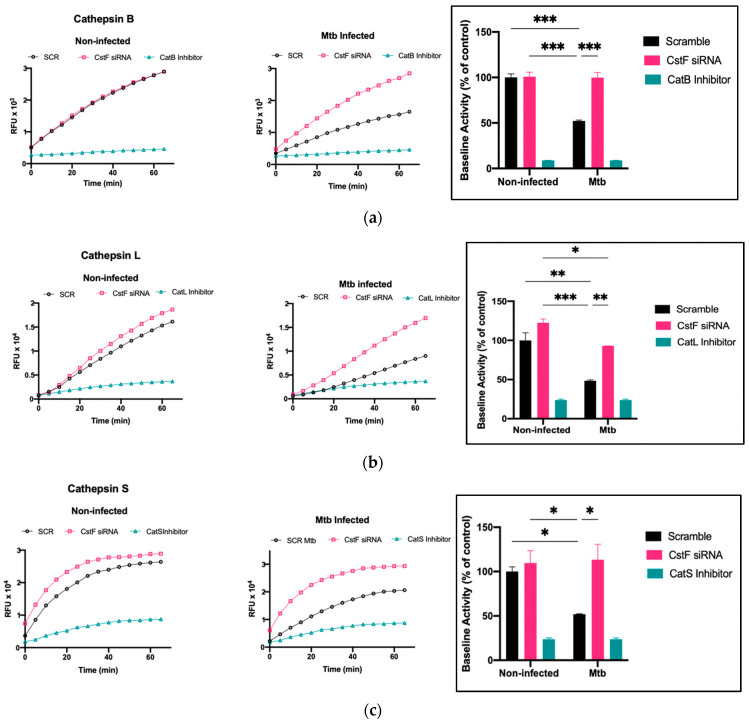
Silencing of CstF expression alters the proteolytic activity of cathepsins in human macrophages infected with Mtb. (**a**) Cathepsin B, (**b**) cathepsin L, and (**c**) cathepsin S activity was measured in scramble control and CstF-silenced cells using cathepsin-specific fluorogenic substrate every 5 min for 60 min. Specific inhibitors were used as negative controls. The bar plots represent average baseline activity calculated as the largest slope of fluorescence emission over 1 h. The slope of fluorescence emission in the control (scramble) for non-infected cells was represented as 100%, and each sample’s effect was shown in a percentage relative to the control. The error bars represent the standard error of the mean. The line plots demonstrate average fluorescence per time. All cathepsin-specific inhibitor controls produced statistically significant inhibition of proteolytic activity (*p* < 0.001). * *p* ≤ 0.05, ** *p* ≤ 0.01, *** *p* ≤ 0.001. Mtb, *Mycobacterium tuberculosis*; RFU, relative fluorescence units.

## Data Availability

Not applicable.

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
