# Peer review of "Modulation of Cystatin F in Human Macrophages Impacts Cathepsin-Driven Killing of Multidrug-Resistant Mycobacterium tuberculosis"

_microorganisms, 2023, doi:10.3390/microorganisms11071861_

Round 1

Reviewer 1 Report

Mtb has extraordinary ability to manipulate the cellular defenses of the host. Mtb evades acidic and proteolytic environment of the phagolysosomes of macrophages, resulting in Mtb growth and multiplications. Cathepsins are endo-lysosomal proteases, play key role in protein degradation, immune response, and host defense against intracellular pathogens. Mtb infection to macrophages increases the expression of cystatin, a natural inhibitor of cathepsins, thereby decreases the activity of cathepsins. In this study, authors have silenced the gene of cystatin F that led to the increase in the activity of cathepsins B, L, and S, resulting in the inhibition of Mtb growth. This observation is true for drug susceptible and drug resistant Mtb, which is quite remarkable and therefore modulation of cystatin F represents a potential target for host directed therapy. However, the study has only in vitro observational data which limits the scope of the study. I also have the following suggestions which could be helpful for the article.

1. This is a small observational study and has 3 figures only. I would suggest including more data.

 2. Figure 2 is all about the infection of macrophages (scramble vs CstF siRNA) with different Mtb strains (laboratory and clinical strains; drug susceptible and drug resistant Mtb). Clinical Mtb strains are more cytotoxic than laboratory Mtb H37Rv. Therefore, this figure should also have the data of macrophage’s death following Mtb infection. It will be good to see if silencing cystatin F increases the proportion of live cells along with reduced Mtb growth.

3. I would suggest including data that shows why cystatin silenced macrophages restrict the Mtb growth better than control macrophages. It will be good if the study has at least one experiment related to bacterial clearance; apoptosis, autophagy, and bacterial clearance via phagolysosomal fusion and this should be figure 4.

4.   Direct evidence is required that shows Mtb growth is impaired in CstF siRNA macrophages that have increased activity of cathepsins. Median fluorescence intensity (MFI) of H37Rv in scramble vs CstF siRNA macrophages obtained from flow cytometry data, will provide direct evidence.

5.       Line 38, it is ten million not then million.

Author Response

Mtb has extraordinary ability to manipulate the cellular defenses of the host. Mtb evades acidic and proteolytic environment of the phagolysosomes of macrophages, resulting in Mtb growth and multiplications. Cathepsins are endo-lysosomal proteases, play key role in protein degradation, immune response, and host defense against intracellular pathogens. Mtb infection to macrophages increases the expression of cystatin, a natural inhibitor of cathepsins, thereby decreases the activity of cathepsins. In this study, authors have silenced the gene of cystatin F that led to the increase in the activity of cathepsins B, L, and S, resulting in the inhibition of Mtb growth. This observation is true for drug susceptible and drug resistant Mtb, which is quite remarkable and therefore modulation of cystatin F represents a potential target for host directed therapy. However, the study has only in vitro observational data which limits the scope of the study. I also have the following suggestions which could be helpful for the article.

  1. This is a small observational study and has 3 figures only. I would suggest including more data.

Re: We hope we have addressed the concerns of the reviewer with the data introduced below in response to different questions.

  1. Figure 2 is all about the infection of macrophages (scramble vs CstF siRNA) with different Mtb strains (laboratory and clinical strains; drug susceptible and drug resistant Mtb). Clinical Mtb strains are more cytotoxic than laboratory Mtb H37Rv. Therefore, this figure should also have the data of macrophage’s death following Mtb infection. It will be good to see if silencing cystatin F increases the proportion of live cells along with reduced Mtb growth.

Re: We reorganized figure 1 to cover all controls related to cystatin F silencing. The figure now includes the western-blot results for protein silencing in addition to gene expression results; the extent of silencing for several days of cell culture and the results showing non-cytotoxic effects on infected cells along the same period.

Figure 2 is related to results in infected cells related to bacterial survival. In addition to intracellular survival, we included the control showing no interferences on internalization due to cystatin F silencing (previously in figure 1) and incorporated the cell death assays that indicate no statistically significant differences on apoptosis/necrosis among reference and clinical strains. We have been using these clinical strains for distinct experiments and we previously adjusted the multiplicity of infection to have similar survival of the infected cells. These results together with those showing no cytotoxic effects of cystatin F silencing substantiate that macrophage death in not contributing to a potential bias of the observed CFU differences.

  1. I would suggest including data that shows why cystatin silenced macrophages restrict the Mtb growth better than control macrophages. It will be good if the study has at least one experiment related to bacterial clearance; apoptosis, autophagy, and bacterial clearance via phagolysosomal fusion and this should be figure 4.

Re: In the present manuscript we intend to demonstrate that the silencing of cystatin F impacts cathepsins hydrolytic activity with consequences on pathogen killing including MDRTB. We have introduced the results of cell death in the different tested strains in figure 2 showing no differences on apoptosis or necrosis. Cystatin F is among cystatins the one that accumulates in higher amounts in endosomal compartments therefore it can only target cathepsins in the endolysosomal pathway and not in the cytosol (as required for apoptotic Bid activation by cathepsin B) or the nucleus in the case of cathepsin L ( https://doi.org/10.3389/fimmu.2022.955407). The results in Figure 3 clearly show the impact of cystatin F silencing on major cathepsins involved in hydrolytic destruction of Mtb. Those cathepsins are also present in endolysosomal compartments of macrophages especially cathepsins S and B. Likewise, cystatin F silencing impacts cathepsin enzymatic activity particularly cathepsin S. While it is described a higher catalytic activity at a pH of 5.5 for cathepsins B and L (https://doi.org/10.1111/j.1600-065X.2007.00552.x) (particularly cathepsin B also have some activity at higher pH) for cathepsin S this lysosomal enzyme possesses a broad pH activity (https://doi.org/10.1016/0076-6879(94)44036-0 ; https://doi.org/10.1515/hsz-2015-0114 ) therefore can be fully activated in any endocytic compartment that holds Mtb from early to late phagosomes or lysosomes just depending on dysfunctional inhibition by cystatins. From the three cathepsins tested cathepsin S is the one with the higher activity during infection in cystatin F silenced conditions as depicted in figure 3. Therefore, experiments showing increase of phagolysosome fusion do not disregard that the killing could have occurred within early phagosomes in those conditions. Altogether our experiments suggest a link between Mtb destruction along the endocytic pathway along with an increase of cathepsin activity, particularly cathepsin S and followed by cathepsin B.

  1.  Direct evidence is required that shows Mtb growth is impaired in CstF siRNA macrophages that have increased activity of cathepsins. Median fluorescence intensity (MFI) of H37Rv in scramble vs CstF siRNA macrophages obtained from flow cytometry data, will provide direct evidence.

Re: To control the accuracy of the CFU experiment we began by showing no differences in pathogen internalization between cystatin F silenced cells relatively to control cells (figure 2a and 2c) and in figure 1 we show no cytotoxic effects induced by cystatin F silencing. In our view, these results support that CFU of recovered intracellular bacteria can be compared between cystatin F silenced conditions relatively to the control. Therefore, the measurement of the median fluorescence intensity would be a redundancy of the CFU counting method, which is the gold standard for bacteria viability. We do understand that additional evidence would further strengthen our conclusions, yet it is our experience that most fluorescent molecules are poor reporters of Mtb survival/death since they accumulate in the macrophage and are still detectable for long periods after the bacteria have been killed.

  1. Line 38, it is ten million not then million.

Re: Thank you for detecting the error, we have now corrected that.

Reviewer 2 Report

Authors found knock-down the expression of cystatin F in macrophage could maintain  cysteine cathepsin enzymatic activity during Mtb infection, and improved pathogen intracellular killing activity of macrophage. The work is  significative and important for MDR and XDR-TB therapy. But I would like to make some suggestions:

1: There are errors in the abstract. "Cathepsins are promising targets due to their manipulation by Mtb with consequences such as decreased proteolytic activity and pathogen survival in macrophages. " Maybe should be "improved its intracellular survival."

2: The decrease expression of cystatin F in macrophage by siRNA should be verified at protein level, such as Wsetern blot.

  •  

Author Response

Authors found knock-down the expression of cystatin F in macrophage could maintain  cysteine cathepsin enzymatic activity during Mtb infection, and improved pathogen intracellular killing activity of macrophage. The work is significative and important for MDR and XDR-TB therapy. But I would like to make some suggestions:

1: There are errors in the abstract. "Cathepsins are promising targets due to their manipulation by Mtb with consequences such as decreased proteolytic activity and pathogen survival in macrophages. " Maybe should be "improved its intracellular survival."

Re: The suggested correction it is now introduced in the abstract.

2: The decrease expression of cystatin F in macrophage by siRNA should be verified at protein level, such as Wsetern blot.

Re: We agree with the reviewer and have now introduced the results of the western blot in figure 1b.

Round 2

Reviewer 1 Report

The article now has new data related to suggestions provided in my first review report. Overall, it is a good study that provides experimental evidence for cystatin F as a potential target for host directed therapy during TB disease.

Minor suggestions

11. Line 20, it should be “inhibitors such as cystatin” in place of “inhibitor as cystatin.”

22. Line 22-25, It will be better if abstract has outcome of the study (increase in the activity of cathepsins, resulting in enhanced killing of Mtb), instead of saying that “cystatin F by siRNA interferes with cathepsins S, L, and B proteolytic activity, significantly impacting pathogen intracellular killing in macrophages.

33. Result, 3.1, line 205, it should be "we investigated" instead of it was investigated the extent of ………….